# Functional Impacts of Epitranscriptomic m^6^A Modification on HIV-1 Infection

**DOI:** 10.3390/v16010127

**Published:** 2024-01-16

**Authors:** Stacia Phillips, Tarun Mishra, Siyu Huang, Li Wu

**Affiliations:** Department of Microbiology and Immunology, Carver College of Medicine, University of Iowa, Iowa City, IA 52242, USA; stacia-phillips@uiowa.edu (S.P.); tarun-mishra@uiowa.edu (T.M.); siyu-huang@uiowa.edu (S.H.)

**Keywords:** epitranscriptomic RNA modification, *N*^6^-methyladenosine, HIV-1, virus–host interactions, innate immunity

## Abstract

Epitranscriptomic RNA modifications play a crucial role in the posttranscriptional regulation of gene expression. *N*^6^-methyladenosine (m^6^A) is the most prevalent internal modification of eukaryotic RNA and plays a pivotal role in RNA fate. RNA m^6^A modification is regulated by a group of cellular proteins, methyltransferases (writers) and demethylases (erasers), which add and remove the methyl group from adenosine, respectively. m^6^A modification is recognized by a group of cellular RNA-binding proteins (readers) that specifically bind to m^6^A-modified RNA, mediating effects on RNA stability, splicing, transport, and translation. The functional significance of m^6^A modification of viral and cellular RNA is an active area of virology research. In this review, we summarize and analyze the current literature on m^6^A modification of HIV-1 RNA, the multifaceted functions of m^6^A in regulating HIV-1 replication, and the role of viral RNA m^6^A modification in evading innate immune responses to infection. Furthermore, we briefly discuss the future directions and therapeutic implications of mechanistic studies of HIV-1 epitranscriptomic modifications.

## 1. Introduction

Post-transcriptional RNA modification is an important mechanism for regulating gene expression. *N*^6^-methyladenosine (m^6^A) is the most prevalent RNA modification present internally within eukaryotic mRNAs and influences RNA stability, translation, splicing, and subcellular localization [1,2,3,4]. Viral RNA has been known to contain m^6^A for nearly 50 years [5]. However, advances in the technology for detecting and mapping sites of modified RNA have resulted in rapid progress in understanding how m^6^A influences virus replication, virus–host interactions, and antiviral immunity [6].

The literature describing m^6^A modification of viral RNA and how m^6^A pathway proteins regulate viral infections has grown at a remarkable pace over the last few years. This review will focus on the current knowledge of how m^6^A modifications influence diverse aspects of the life cycle of human immunodeficiency virus type 1 (HIV-1). We discuss the topology of m^6^A modification on HIV-1 RNA, the roles of m^6^A writers, readers, and erasers in regulating HIV-1 replication, and how m^6^A affects the induction of innate immunity in response to HIV-1 infection.

## 2. Cellular m^6^A-Regulating Proteins and Their Functions

The addition of a methyl group to the *N*^6^ position of adenosine is catalyzed by a multi-subunit methyltransferase complex, also referred to as the m^6^A writer complex. The catalytic core is formed by a heterodimer of methyltransferase-like 3 (METTL3) and methyltransferase-like 14 (METTL14), with Wilms tumor 1-associating protein (WTAP) serving as a scaffold protein that stabilizes the methyltransferase complex [7,8]. Additional components of the methyltransferase complex that may direct the complex to sites of m^6^A modification include RNA-binding motif protein 15 (RBM15) and KIAA1429, also known as vir-like m^6^A methyltransferase-associated (VIRMA) [9,10]. However, the function of each protein associated with the methyltransferase complex is not fully understood.

The writer complex exhibits a preference for modification of adenosine in the consensus sequence 5′-DRACH-3′ (D = A/G/U, R = A/G, H = U/A/C) in mammalian mRNAs [11]. Transcriptome-wide m^6^A mapping techniques have revealed that m^6^A is most common in the 3′ UTRs and long exons of mRNA [11]. Although DRACH motifs occur frequently, only ~10% of them are m^6^A-methylated in cells, suggesting that there are context-dependent signals for the methylation of specific DRACH motifs [11].

m^6^A modification is reversible through the action of demethylases, or erasers. These enzymes include the alkB homolog 5 (ALKBH5) and fat mass and obesity-associated protein (FTO), and both enzymes can remove m^6^A modification [12,13]. However, since the discovery of these enzymes, it has been shown that FTO exhibits a strong preference for demethylating *N*^6^, 2′-O-methyladenosine (m^6^Am) residues next to the 5′ cap, rather than m^6^A at internal sites of mRNA [14]. Therefore, while the expression of individual eraser enzymes can be manipulated to affect overall changes in m^6^A levels, the role of each endogenous eraser in the context of virus infection in cells remains unclear.

The functional consequence of m^6^A modification is determined by the binding of m^6^A-specific RNA-binding proteins, or readers. Readers that are most often studied in the context of m^6^A function are the YTH domain-containing family of proteins, YTHDC1 and YTHDF1–3. YTHDC1 regulates the nuclear export of mature mRNA [3]. The cytoplasmic YTHDF1–3 proteins were initially determined to be functionally distinct. YTHDF1 promotes translation, while YTHDF2 mediates the degradation of m^6^A-modified RNAs [1]. YTHDF3 acts in synergy with YTHDF1 and YTHDF2 to enhance translation or mediate RNA decay [15]. However, more recent studies suggest that YTHDF1–3 are functionally redundant and mediate RNA degradation [16]. The discrepancies regarding YTHDF protein functions have been reviewed elsewhere [17,18,19]. While YTHD family readers are often the focus of studies designed to elucidate the role of m^6^A in viral replication, many other potential reader proteins have also been described (reviewed in [17]). These less-studied m^6^A readers also deserve attention in investigating the mechanisms of m^6^A in regulating cellular gene expression and virus replication. 

A major challenge in determining the function of m^6^A in regulating virus replication is the ability to specifically remove m^6^A on viral RNA, especially transcripts, while leaving the modification of cellular RNA unperturbed. Almost all studies to date have utilized overexpression or knockdown/knockout of writers, readers, and erasers to infer the function of viral RNA m^6^A during infection. However, these approaches will also lead to changes in the m^6^A modification of host cell RNA, which may have indirect effects on virus replication. Such indirect effects must always be carefully considered during the interpretation of these studies. Additional approaches, including mutations of m^6^A sites in viral genomes, are often used in confirming specific effects [20,21,22,23,24].

## 3. Mapping m^6^A Modification Sites on HIV-1 RNA

Several studies have reported the location of m^6^A in HIV-1 RNA from infected cells or purified virions (Figure 1A,B). The virus strains, cell types, and sequencing methods used in studies are summarized in Table 1. To date, all HIV-1 m^6^A-mapping studies have used CXCR4-tropic viruses (LAI and NL4-3 strains). In the first report, methylated RNA immunoprecipitation sequencing (MeRIP-seq) was used to determine m^6^A-modified regions of HIV-1 RNA from infected MT4 CD4+ T cells [25]. meRIP-seq utilizes fragmented RNA for m^6^A-specific immunoprecipitation followed by high-throughput sequencing and has a resolution of ~100–200 nt [11]. This analysis identified 14 regions predicted to contain m^6^A modifications [25] (Figure 1A). In contrast to subsequent reports, this data analysis did not include the 5′ and 3′ UTR sequences, which led to the absence of input or m^6^A signal mapping to these regions. Peaks were mapped primarily to coding sequences and exhibited overlap with several splicing regulatory sequences. Unique to this study was the identification of a peak located in the *env* and *rev* response element (RRE), which was the focus of subsequent functional analysis. Methylation of predicted sites in the RRE was confirmed using a primer extension assay in infected cells [25]. However, none of the subsequent studies reported by other groups found m^6^A to be present in the RRE [26,27].

The next report identified sites of m^6^A modification in HIV-1 RNA from CEM-SS CD4+ T cells infected with single-cycle HIV-1 [26]. This study employed photo-crosslinking-assisted m^6^A sequencing (PA-m^6^A-seq), which modifies the meRIP protocol through the addition of 4-thiouridine (4SU) labeling of viral RNA followed by UV crosslinking of an m^6^A-specific antibody to the labeled RNA prior to sequencing. This approach improves the resolution of sites of m^6^A modification to ~20 nt due to the introduction of U-to-C mutations at the sites of crosslink between the antibody and 4SU [32]. In this analysis, all predicted sites of methylation were mapped to the 3′ ~15% of the genome (Figure 1A). The PA-m^6^A-seq results were compared to photoactivatable ribonucleoside-enhanced crosslinking and immunoprecipitation (PAR-CLIP) performed to identify the binding sites for YTHDF reader proteins on the HIV-1 RNA. Comparison of reader-binding sites in HIV-1 RNA from HEK293T cells transfected with an HIV-1 proviral plasmid and m^6^A peaks from infected CEM-SS cells revealed four regions in common, located in *env/rev*, *nef*, and the 3′ UTR. The authors also sought to assess the topology of reader-binding sites in different HIV-1 strains. PAR-CLIP data assessing reader-binding sites in the RNA of HIV-1_BaL_ and HIV-1_JR-CSF_ (both are R5-tropic) show that these primary isolates have the four binding sites in common with HIV-1_NL4-3_, while also possessing additional reader-binding peaks. This illustrates the importance of considering cell type- and HIV-1 strain-specific differences in m^6^A modification of viral RNA. This group also reported the location of m^6^A modifications in HIV-1 genomic RNA (gRNA) purified from virions compared to cell-associated RNA from CEM-SS cells. PA-m^6^A-seq revealed major peaks in *env/rev*, *nef*, and the 3′ UTR of gRNA, consistent with the results of m^6^A mapping to cell-associated viral RNA (Figure 1B) [28]. Interestingly, additional peaks were detected in several coding sequences of virion RNA including *pol*, *env*, and *vpr*. 

**Figure 1 viruses-16-00127-f001:**
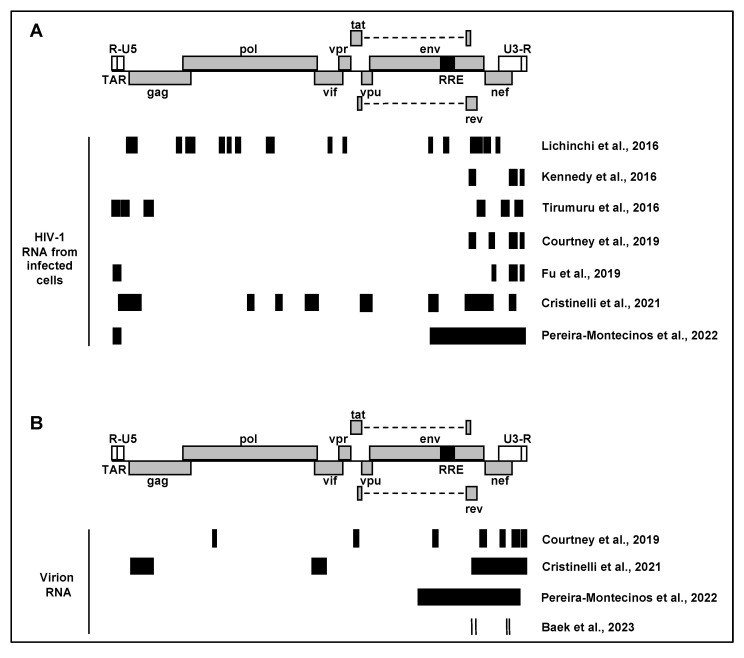
Summary of m^6^A mapping in HIV-1 RNA. The position of the m^6^A peaks on the HIV-1 genome is illustrated by a schematic diagram that is drawn to scale. The HIV-1_NL4-3_ strain genome was used as a reference to compare data reported by each publication. Black boxes represent predicted peak locations based on various sequencing approaches using RNA purified from (**A**) HIV-1-infected cells or (**B**) HIV-1 virions. Where possible, boxes represent exact reported peak width [25,26,27,28,31,33]. Other boxes are drawn based on peak calling maps from individual publications [29,30].

Another study using meRIP-seq confirmed the presence of major m^6^A peaks in *env/rev*, *nef*, and the 3′ UTR in HIV-1 RNA from infected Jurkat and primary CD4+ T cells (Figure 1A) [27]. This analysis also identified additional smaller but prominent peaks in the 5′ UTR and *gag*. Cell-associated HIV-1 RNA collected from HEK293T cells transfected with an HIV-1 proviral plasmid showed similar sites of m^6^A modification as viral RNA from infected CD4+ T cells, with an additional major peak in *tat*. These results suggest that in addition to cell type differences, it is possible that patterns of HIV-1 m^6^A modification differ in virus-producer cells and target cells. Validation of m^6^A modification at the predicted sites was performed by crosslinking–immunoprecipitation sequencing (CLIP-seq) to identify the binding sites of YTHDF1–3 proteins on HIV-1 RNA isolated from infected HeLa/CD4 cells. CLIP-seq analysis revealed reader-binding sites in *env*/*rev*, *nef*, and the 5′ UTR, providing further support for the presence of m^6^A at these locations. Of note, reader binding was detected in several coding regions of the HIV-1 genome, sites that were not detected by meRIP-seq. This could be due to different cell types used for meRIP-seq and CLIP-seq or due to more sensitive and robust detection using a crosslinking approach.

Mapping sites of m^6^A modification at a single timepoint does not provide information about how modification states may change over the course of infection. One group investigated whether temporal changes in m^6^A modification of viral RNA occur at different times during a single-cycle HIV-1 infection [29]. meRIP-seq was performed using cell-associated RNA from SupT1 cells infected with single-cycle HIV-1 for 12, 24, or 36 h post-infection (hpi). Each of these time points is earlier than the previously reported analyses, all performed at 96 hpi. m^6^A peak mapping at these earlier time points again showed enrichment of m^6^A in the 3′ end of the RNA, with peaks also detected in *pol*, *pol/vif*, and *vpu* (Figure 1A). One notable change in m^6^A mapping over time revealed a large peak overlapping the RNA-packaging signal ψ at 36 hpi but not earlier. RNA isolated from virions released from the same cells at 36 hpi retained m^6^A modification in the *env/nef* and 3′ UTR regions, as well at the peak overlapping ψ, but was largely lacking prominent peaks in the remaining coding regions (Figure 1B). These results suggest that m^6^A modification patterns differ between packaged gRNA and translated viral transcripts. The peak width defined for these analyses was quite broad and ranged from ~200–500 nt, while meRIP-seq involves fragmentation of RNA to 100–200 nt in length. This low resolution prevents meaningful prediction of specific DRACH motifs that may be modified. 

Another report comparing m^6^A modification of infected-cell HIV-1 RNA to RNA purified from viral particles also showed that the cell-associated RNA has a different pattern of methylation than that of packaged gRNA [30]. RNA was prepared from HEK293T cells transfected with HIV-1 proviral plasmids or from supernatant virus. Consistent with all other reports, both viral RNA preparations exhibited peaks overlapping in *env/rev*, *nef*, and the 3′ UTR (Figure 1A,B). In contrast to the results reported by Cristinelli et al. [29], a peak was observed in the 5′ UTR of intracellular RNA that was absent from virion RNA. The authors concluded that m^6^A in the 5′ UTR regulates genome packaging and focused on this phenotype for subsequent functional studies. 

Techniques such as meRIP-seq and PA-m^6^A-seq reveal the population-level modification state of fragmented RNA and therefore cannot provide information on the presence of m^6^A in individual viral genomes or transcripts. Therefore, these data do not provide insight into whether there is differential m^6^A deposition among specific splice isoforms. The most recent analysis of m^6^A sites in HIV-1 RNA was performed using a modification of direct RNA sequencing (DRS) that resulted in a remarkable improvement in full-length genome reads from 0.01% to 34.9% [31]. This approach allowed for full-length, single-molecule analysis at single-nucleotide resolution for both viral gRNA and splice isoforms. Additional stringency was provided by using three independent bioinformatics tools for the identification of m^6^A signals. Seven common sites were identified across all three analyses. Interestingly, two of these sites, both in *pol*, were not located in canonical DRACH motif sequences (Figure 1A). Four sites in the 1.2 kb of the 3′ HIV-1 genome coincide with the common peaks identified in *env*/*rev* and the 3′ UTR in each of the previous studies [26,27,28,29]. These m^6^A sites are located at positions 8079, 8110, 8975, and 8989 of HIV-1_NL4-3_ [31]. 

Almost all available data describing the location of m^6^A in HIV-1 RNA are derived from cell lines infected *in vitro*. One study demonstrated that the pattern of viral RNA modification is similar in a CD4+ T cell line and primary CD4+ T cells infected *ex vivo*, suggesting that these results might be relevant to HIV-1-infected individuals [27]. The only information regarding *in vivo* HIV-1 infection comes from a transgenic rat model of HIV-1 infection [33]. These animals harbor a *gag*/*pol*-deleted provirus that expresses viral genes in microglia and astrocytes [34]. meRIP-seq of RNA isolated from the hippocampus of these animals revealed m^6^A in the 5′ UTR and *nef* and 3′UTR, consistent with cells infected *in vitro* and *ex vivo* (Figure 1A) [27,33].

Differences in m^6^A site prediction among studies can arise for several reasons, including biological, technical, and computational differences in the experimental approach. Such variables include HIV-1 strain, cell type, virus purification method, sequencing approach, data processing, and bioinformatic tools. In addition to the issue of low resolution provided by meRIP-seq, concerns have been raised about the reproducibility of the method, even when using the same experimental conditions [35]. Future studies would benefit from the use of sequencing approaches that allow for the identification of m^6^A at single-nucleotide resolution.

Despite some inconsistencies among studies, m^6^A sites located in *env* (two sites) and the 3′ UTR (two sites) have been reproducibly identified across several data sets. In addition, adenosine residues in viral DRACH motifs that are predicted to contain m^6^A are highly conserved in the HIV-1 genome that is otherwise highly susceptible to mutation [25,26,31,36]. In addition, only a very small fraction of the nearly 250 DRACH motifs in the full-length HIV-1 RNA [31] appear to contain m^6^A, suggesting that modification of these sites is very highly specific. These data suggest that these m^6^A modifications are especially important in facilitating efficient virus replication. 

## 4. HIV-1 Infection Modulates the Cellular RNA m^6^A Profile

Epitranscriptomic modification of mRNA represents an important layer of gene expression regulation in mammalian cells. While much focus has been placed on determining the m^6^A profile of HIV-1 RNA, another area of interest is how viral infection influences changes in the m^6^A modification of host cell transcripts. This can provide mechanistic insights into how HIV-1 induces changes in host cell gene expression, either directly through the action of viral proteins or because of the intrinsic antiviral response to infection.

Infection of various cell types including differentiated monocytic, microglial, and CD4+ T cell lines, as well as primary CD4+ T cells, with HIV-1 induces an upregulation of m^6^A levels in total cellular RNA through mechanisms that remain unclear [25,29,37,38,39]. Upregulation of m^6^A in CD4+ T cells and microglial cells is not due to changes in the expression of m^6^A writers or erasers [37]. It is possible that methyltransferase complex activity is being regulated at the level of complex formation, stability, or enzymatic activity.

In CD4+ T cells, m^6^A upregulation is also observed in cells treated with the reverse transcriptase inhibitor nevirapine, heat-inactivated virus, and gp120, the surface unit of the HIV-1 envelope protein [37]. These data demonstrate that m^6^A upregulation is independent of virus replication. m^6^A upregulation occurs after treatment of cells with recombinant gp120 proteins from CXCR4- and CCR5-tropic HIV-1 and is blocked by occlusion of the primary receptor CD4 with anti-CD4 antibodies [37]. Whether HIV-1 co-receptor binding to gp120 is also required for m^6^A upregulation of cellular RNA remains unknown. Analysis of m^6^A levels in polyadenylated and non-coding RNA reveals an upregulation of m^6^A in both, including ribosomal RNA (rRNA) [37]. The 18S and 28S rRNA subunits both contain a single m^6^A that is modified by the methyltransferases METTL5 and ZCCHC4, respectively [40,41]. Ribosomes will form normally in the absence of METTL5 or ZCCHC4, but the overall translational efficiency is reduced [40,42]. This raises the interesting possibility that HIV-1 may enhance its protein translation through m^6^A modification of the ribosome; however, this possible mechanism has not yet been explored.

High throughput sequencing approaches such as meRIP-seq allow for the identification of changes in the relative abundance of m^6^A-modified host cell mRNAs. The first transcriptome-wide meRIP-seq defining the topology of m^6^A in human cells showed an enrichment of m^6^A near stop codons and in coding sequences, particularly long exons [11]. Mapping of m^6^A in cellular transcripts from HIV-1-infected CD4+ T cell lines compared to uninfected controls reveals no significant changes in the distribution of m^6^A deposition to the 5′ UTR, exons, introns, or the 3′ UTR, with the majority of m^6^A sites found in exons and the 3′ UTR as expected [25,27,29]. 

Changes in the relative abundance of m^6^A-modified transcripts alone do not indicate differential methylation of the RNA, because these changes could simply be a result of differential expression of m^6^A-modified transcripts. Therefore, true differential m^6^A methylation of HIV-1 infected cells can only be determined by measuring the modification level and transcript abundance and comparing those data to the same transcript in uninfected control cells. By applying this strategy, over 3615 transcripts were found to be hypermethylated upon HIV-1 infection compared to hypomethylation of only 777 transcripts [29]. These data support an overall upregulation of m^6^A modification of cellular transcripts in SupT1 cells upon HIV-1 infection. Comparing m^6^A methylation patterns at 12, 24, and 36 h post-infection, only 2% of differentially methylated transcripts were common among all three time points [29]. This shows that the m^6^A modification patterns of cellular RNA are dynamic and temporally regulated during HIV-1 infection.

A new analysis combining both RNA-seq and meRIP-seq data showed that changes in transcript methylation occurred disproportionately in coding sequences compared to the 3′ UTR in a microglial cell line during a single-cycle HIV-1 infection [38]. Over 2000 transcripts were reported to be hypo- or hyper-methylated in infected cells, suggesting that HIV-1 infection induces changes in gene expression at the level of m^6^A modification. Pathway analysis of these transcripts showed significant enrichment of genes involved in signal transduction, including the Ras and MAPK signaling pathways [38]. Further studies will be required to determine the functional effect of differential methylation on these signaling pathways during HIV-1 infection of microglia.

## 5. HIV-1 m^6^A Suppresses the Induction of Type I Interferon (IFN-I) in Macrophages

CD4+ T cells are highly permissive to HIV-1 infection and are rapidly killed by active replication [43], while macrophages are infected at much lower levels and do not exhibit the cytopathic effect that is typical for CD4+ T cells [44]. As innate immune cells infected with HIV-1, macrophages are thought to play diametrical roles during HIV-1 infection. Infected macrophages can contribute to virus dissemination and persistence, and reactivation of the latent reservoir [45,46]. In contrast, as innate immune surveillants, macrophages are critical for mounting an antiviral immune response during viral infections. Therefore, it is important to understand how HIV-1 inhibits IFN-I during the infection of macrophages, allowing for the establishment of latent infection. Internal 2′-O-methylation of HIV-1 RNA serves as an immune evasion strategy during HIV-1 infection of monocyte-derived macrophages and dendritic cells [47]. To test whether m^6^A also suppresses IFN-I in response to HIV-1 infection, viruses were produced in the presence of eraser (ALKBH5 or FTO) overexpression or knockout [48]. Differentiated monocytic U937 cells (macrophage-like cells) were then infected with HIV-1 containing low, normal, or high levels of m^6^A in the incoming RNA genome. The important advantage of manipulating m^6^A levels in producer cells rather than target cells is that it prevents the perturbation of cellular m^6^A levels so that observed phenotypes can be attributed specifically to m^6^A in the incoming HIV-1 gRNA. Using complementary approaches for modulating m^6^A, the authors demonstrated that the level of m^6^A in the HIV-1 genome is inversely correlated with the expression of IFN-α and IFN-β after transfection or infection of differentiated U937 cells [48]. The phenotype was observed in primary monocyte-derived macrophages transfected with HIV-1 RNA [48]. Taken together, the results suggest that m^6^A suppresses the expression of IFN-I by HIV-1 RNA. Delivery of RNA to cells by transfection does not recapitulate the pathway by which gRNA is delivered to cells during infection. Therefore, future mechanistic studies should be conducted using HIV-1-infected cells, particularly macrophages.

Viral RNA is detected in the host cell by protein sensors that recognize pathogen-associated molecular patterns in viral molecules to discriminate self from non-self RNA [49]. The binding of these sensors to their RNA ligands activates a signaling cascade that induces the expression of IFN-I. To determine which host cell cytosolic RNA sensor may be inhibited by m^6^A-modified HIV-1 RNA, U937-knockout (KO) or -knockdown (KD) cell lines were generated lacking the expression of retinoic acid-induced gene I (RIG-I) or melanoma differentiation-associated gene 5 (MDA5), respectively. These cells were transfected with a 42-mer HIV-1-derived RNA oligonucleotide with or without a single m^6^A. The sequence of the oligonucleotide corresponds to a portion of the HIV-1 5′ UTR of HIV-1, where potential m^6^A sites were predicted by meRIP-seq studies [25,27]. While the expression of IFN-α and IFN-β was lower in both RIG-I KO and MDA5 KD cells compared to control, RIG-I KO cells were no longer able to discriminate between oligonucleotides with or without m^6^A [48]. These data suggest that m^6^A may function by evading RIG-I sensing, and this would be consistent with a growing body of research in the field of innate immunity to RNA viruses [50,51,52,53]. Also, it was shown that the infection of differentiated U937 cells with m^6^A-deficient HIV-1 induces more phosphorylation of IRF3 and IRF7 as compared to control viral particles. Based on this, it appears that the presence of m^6^A on HIV-1 RNA hinders IFN-1 production during the early stages of HIV-1 infection. However, a major limitation of this study is the use of RNA oligonucleotide transfection, especially given uncertainties regarding the exposure of HIV-1 genomes to the cytoplasm during transit to the nuclear pore [54,55,56,57]. It will be critical to determine whether m^6^A in the HIV-1 genome avoids detection by RIG-I in the context of infected cells, and if so, what are the underlying mechanisms. It is possible that other cellular RNA sensors might be involved in sensing HIV-1 RNA with lower levels of m^6^A.

## 6. m^6^A Reader Proteins Negatively Impact HIV-1 Reverse Transcription

The sequencing approaches described above demonstrate that m^6^A is present in viral genomes and transcripts. Therefore, m^6^A may impact both pre- and post-integration events in HIV-1 replication. Tirumuru et al. reported that YTHDF proteins have a negative impact on post-entry viral replication in several target cell types, including primary CD4+ T cells [27]. After virus entry, the measurable level of *gag* RNA drops relative to input, reflecting the degradation of incoming genomes during reverse transcription (RT). The levels of *gag* RNA then increase after the integration and transcription of new viral RNA from the provirus. However, *gag* RNA levels fail to increase in cells constitutively overexpressing YTHDF proteins, which is consistent with the degradation of gRNA and a concomitant decrease in early and late RT products due to the lack of RT template [27,58]. Both cell-based and *in vitro* binding assays demonstrated the binding of YTHDF proteins to HIV-1 RNA [27,58]. However, these binding assays were performed *in vitro* or post-lysis of infected cells. Therefore, it is not clear whether the HIV-1 genome and transcripts interact with all three YTHDF proteins in cells during viral infection. 

There are opposing results regarding whether YTHDF reader proteins are incorporated into HIV-1 particles. One study shows that both overexpressed and endogenous YTHDF3 are incorporated into virions in a nucleocapsid-dependent manner in both HEK293T cells transfected with HIV-1 proviral plasmids and infected A3R5-Rev-GFP cells (HIV Rev-dependent reporter CD4+ T cells expressing CCR5), but the incorporated reader is then degraded by the viral protease [59]. In contrast, two other studies, both using highly purified virions, reported no detectable incorporation of FLAG-tagged YTHDF1–3 proteins into virus particles prepared from HEK293T cells transfected with HIV-1 proviral plasmids [58,60]. In one of these latter studies, YTHDF1–3 were not detected in virions even when HIV-1 protease was inhibited [60]. The reason for this discrepancy could be due to differences in HIV-1 purification methods. Future studies may address whether HIV-1 RNA is degraded in the presence of endogenous YTHDF proteins in CD4+ T cells or macrophages, and the underlying mechanisms.

## 7. m^6^A Regulates HIV-1 RNA Splicing and Nuclear Export

HIV-1 gene expression is driven by the viral 5′ long terminal repeat (LTR) of integrated provirus in a productively infected cell. The deposition of m^6^A on nascent transcripts occurs co-transcriptionally in the nucleus [61]. It is therefore possible for m^6^A to affect RNA splicing and nuclear export. The first study reporting the presence of m^6^A in HIV-1 RNA identified m^6^A enrichment in the RRE, with two potential methylation sites in hairpin loop IIB [25]. The RRE is a highly conserved and structured RNA element in HIV-1 RNA that is essential for the nuclear export of partially spliced or unspliced transcripts [62]. The authors hypothesized that m^6^A within the RRE may be required for efficient viral RNA export. Mutation of the predicted methylation site in the hairpin IIB bulge dramatically reduced the levels of *env* mRNA in HEK293T cells transfected with HIV-1 proviral DNA plasmids. Subcellular fractionation prior to measurement of *env* mRNA indicated a defect in nuclear export. However, Rev binding to these mutated RREs was not measured. A variety of *in vitro* assays assessing the structure and Rev-binding properties of synthesized RRE IIB with or without m^6^A argued against a significant impact of m^6^A modification on the IIB structure [63]. Further, a separate study made the same RRE mutations in a different HIV-1 laboratory strain and showed that when *env* and *rev* are provided *in trans*, there is no difference in viral gene expression in either producer or target cells [36]. The RRE was not identified as a site of m^6^A modification or reader binding in any subsequent studies [26,27,28]. Therefore, the function of m^6^A in the RRE remains to be confirmed.

The nuclear m^6^A reader YTHDC1 is known to be involved in RNA stability and splicing and the nuclear export of m^6^A-modified viral and cellular RNA [3,64,65]. Two independent studies assessed HIV-1 RNA splicing and nuclear export under conditions of overexpression or depletion of YTHDC1 during single-round infections [26,66]. Both studies found that YTHDC1 is required for not only maintaining overall viral RNA levels but also for the appropriate selection of splice sites. Splicing of HIV-1 transcripts is complex and involves the utilization of multiple splice donor and acceptor sites [67]. Specifically, YTHDC1 increased the use of 3′ splice acceptor sites with a concomitant decrease in the use of splice acceptor sites further 5′ in the HIV-1 genome [60]. Manipulating the expression levels of YTHDC1 had no effect on the cellular NONO mRNA that is not m^6^A-modified [60]. This suggests that the observed changes in HIV-1 RNA upon YTHDC1 silencing or overexpression are due to the presence of m^6^A (Figure 2). Overexpressed and endogenous YTHDC1 bind to spliced and unspliced HIV-1 RNA in infected cells in a METTL3-dependent manner [66] (Figure 2). YTHDC1-binding sites were identified adjacent to HIV-1 splice acceptor (SA) sites 3 and 7 [60]. However, when the predicted m^6^A site near SA3 was mutated, there was no effect on the utilization of SA3 [60]. Therefore, whether YTHDC1 is acting directly on viral m^6^A is unclear. Both groups also reported that overexpression or depletion of YTHDC1 had no effect on the nuclear/cytoplasmic accumulation of unspliced, partially spliced, or completely spliced viral RNA [60,66]. These results argue against a role of YTHDC1 in HIV-1 viral RNA nuclear export, which is unexpected given the role of YTHDC1 in the nuclear export of cellular mRNA [3]. 

Several independent studies reported m^6^A modification in the 3′ region of HIV-1 RNA to overlap between *env* and the second exon of *rev* [26,27,28,29,30,33]. DRS identified the modified adenosine in a DRACH motif in the *env/rev* overlap at position 8079 of HIV-1_NL4-3_, downstream of SA7 [31]. Two other high-confidence m^6^A sites are located in the 3′ UTR. These m^6^A sites represent three of the four most prominent sites of m^6^A modification, in agreement with other studies, and were mutated for functional studies. Interestingly, mutation of these sites individually had no significant effect on viral protein expression, the proportion of unspliced RNA, p24 release, or virion infectivity [31]. Only when all three adenosines were mutated was there a significant reduction in virus infectivity. The reduction in HIV-1 replication is likely due to a decrease in the abundance of unspliced RNA, which is necessary to produce infectious virions. The triple mutation also results in modest but significant increases in the proportion of partially and completely spliced viral RNA relative to total RNA [31]. This suggests functional redundancy of the individual m^6^A and that the modifications are involved in the regulation of splicing. This idea is further supported by the observation that completely spliced RNAs are more m^6^A-modified at these three predominant sites than unspliced RNAs [31]. It will be interesting to uncover the mechanisms by which these three m^6^A sites modulate HIV-1 RNA splicing, whether other m^6^A sites are also functionally redundant, and how a single nucleotide modification can substitute for the function of another.

## 8. m^6^A Enhances Post-Integration HIV-1 RNA Abundance, Stability, and Translation

One strategy to indirectly test whether m^6^A inhibits or enhances HIV-1 replication is to manipulate the expression of writer complex components or m^6^A erasers in target cells. Silencing or overexpression of writers or erasers would presumably only affect viral RNA that is made post-integration; however, this has not been assessed in detail. Several studies that have used this approach agree that in general, m^6^A is required for efficient HIV-1 replication as measured by viral RNA, viral protein, and progeny particle release [25,27,30,60]. One exception to this general phenotype is a study using single-cycle HIV-1 transduction of HeLa cells [66]. The exact reason for the discrepancy is unknown but could be due to the kinetics of viral gene expression in this experimental system compared to replication-competent HIV-1 entry by plasma membrane fusion. 

Another area of controversy in the field of HIV-1 and m^6^A is the effect of readers on viral replication. One study measured HIV-1 RNA and protein levels in HEK293T producer cells overexpressing each YTHDF1–3 reader individually and found that in all cases, both HIV-1 RNA and protein levels were increased [26] (Figure 2). In addition, silencing of YTHDF2 decreases the half-life of HIV-1 RNA and cellular RNAs containing m^6^A yet has no significant effect on cellular RNAs that are not modified [60]. These results all suggest that m^6^A readers enhance HIV-1 replication by stabilizing viral RNA. The overexpressed readers are assumed to be acting on viral RNA; however, enhanced replication can also be due to changes in cellular gene expression that indirectly affect viral gene expression. Consistent results were reported after YTHDF2 overexpression or knockout in CEM T cells infected with single-cycle HIV-1 [26,60]. In these cells, there was a reduction in Gag protein levels and p24 levels of released virions, again suggesting that YTHDF2 is required for efficient viral gene expression (Figure 2). These results are in direct opposition to other reports that show a decrease in viral RNA and protein in infected HeLa, Jurkat, and primary CD4+ T cells [27,58]. As discussed above, this was concluded to be a result of the inhibition of RT through the binding of readers to incoming viral RNA, leading ultimately to its degradation [27,58,59]. One possibility for these seemingly opposing results is that m^6^A has a negative effect on HIV-1 gRNA but a potential benefit at the post-integration phases of infection. The use of different cell types, viruses, and lengths of infection make direct comparisons of data from different groups imprecise.

Two potential functions for viral RNA m^6^A are destabilization of RNA or enhanced translation efficiency mediated by m^6^A readers [1,2]. To assess the functional significance of m^6^A modifications in the 3′-UTR of HIV-1 RNA, Kennedy et al. placed the wild-type or m^6^A-deficient (A-to-G mutated) forms of the HIV-1 3′-UTR downstream of a reporter gene [26]. Their results revealed that substituting A with G significantly reduced both the RNA and protein levels of the reporter. These data argue against RNA destabilization or enhanced translation and indicate that m^6^A in the 3′ UTR of HIV-1 RNA enhances the abundance of an unrelated RNA (Figure 2). Recruiting any of the YTHDF readers to the 3′UTR of a reporter, independent of m^6^A, also resulted in enhanced expression of the reporter [26]. Taken together, these results suggest that m^6^A modification in the 3′UTR of HIV-1 mRNA enhances viral gene expression at the mRNA and protein levels.

## 9. m^6^A Inhibits HIV-1 RNA Packaging and Reduces Virion Infectivity

Several groups reported the presence of m^6^A in the 5′ UTR of HIV-1 RNA [27,29,30]. The 5′ UTR is highly structured and contains several regulatory elements that are indispensable for efficient HIV-1 replication [68]. One important aspect of HIV-1 replication that is mediated by RNA sequences in the 5′ UTR is the selection of full-length RNA genomes for packaging into progeny virions [69]. There are two DRACH motifs of particular interest in the 5′UTR. One is in the primer-binding site (PBS) required for the initiation of RT and another is near the dimer initiation sequence (DIS) that is involved in a monomer–dimer switch that regulates whether a given full-length HIV-1 RNA is translated (monomer) or selected for packaging (dimer) [58]. 

Mutation of either of these two adenosine residues led to a slight increase in p24 levels in HEK293T producer cells yet reduced the infectivity of the released virions when used to infect target cells [58]. RNA secondary structure prediction in silico showed an altered structure of the PBS but not the DIS because of the mutations [58]. However, due to the possibility of the disruption of primer binding for RT or Gag binding during particle assembly as a result of these mutations, the interpretation of these data with respect to m^6^A modification can be multifaceted.

Several lines of evidence suggest that m^6^A modification impairs the packaging of gRNA. meRIP-seq data derived from intracellular viral RNA and virion RNA revealed a prominent peak representing m^6^A in the 5′ UTR of intracellular RNA but not RNA from purified virus particles [30]. In addition, the m^6^A/A level of intracellular viral RNA was significantly higher than that in packaged RNA [30]. This raises the possibility that m^6^A plays a role in the preferential selection of gRNA molecules for packaging. Overexpression of METTL3/14, which presumably increases m^6^A modification of viral RNA, resulted in higher levels of Gag protein and lower levels of RNA packaging, whereas silencing of the methyltransferase complex resulted in the opposite phenotype [30]. These results are consistent with the observed decrease in Gag protein in other studies where METTL3 is silenced [25,27]. Interestingly, the interaction between Gag and full-length viral RNA in the cytoplasm was decreased under conditions of METTL3/14 overexpression, further supporting the idea that increased methylation of viral RNA inhibits its ability to be packaged [30]. Unexpectedly, the deletion of the two adenosine residues that are potential sites of methylation resulted in a significant increase in the overall m^6^A content of viral RNA and a significant decrease in the packaging efficiency of gRNA [30]. Due to the deletion rather than mutation of these adenosines, it is unknown whether the proper secondary structure of the 5′ UTR was affected. Another study also demonstrated that virion RNA contains fewer m^6^A modifications than viral transcripts; however, this study only reported modification of three adenosines in the 3′ ~1.4 kb of the viral genome. In addition, YTHDF proteins bind to m^6^A-modified gRNA and form a complex with Gag in an RNA-dependent manner [27,58]. It is therefore possible that YTHDF binding prevents efficient binding of Gag and therefore inhibits packaging. Further experiments are needed to investigate this hypothesis.

HIV-1 Gag colocalizes in the nucleus with unspliced viral RNA [70]. Interestingly, Gag also colocalizes with FTO exclusively in the nucleus and there is an increase in viral RNA m^6^A levels in the absence of gag expression that can be rescued when Gag is provided *in trans* [30]. These data suggest that FTO mediates demethylation of viral RNA in a Gag-dependent manner (Figure 2). However, while ALKBH5 and FTO can both demethylate HIV-1 RNA when each demethylase is overexpressed, it is not known which of these proteins acts on viral RNA under conditions of endogenous expression levels. Emerging evidence supports a predominant role for FTO in the removal of methyl groups from m^6^Am and not m^6^A in human cells [71,72]. Further studies are needed to determine the molecular mechanisms of the removal of m^6^A from HIV-1 RNA and how this process is regulated to control the translation or packaging of full-length gRNA.

## 10. m^6^A Modulation May Have Therapeutic Potential for HIV-1-Infected Individuals

Overall, it is clear that m^6^A modification of HIV-1 RNA is required for proper regulation of viral gene expression, although the molecular mechanisms remain to be elucidated. This raises the question of whether m^6^A deposition is a candidate target for anti-viral therapy or for new strategies toward HIV-1 functional cure. Indeed, the methylation inhibitor 3-deaza-adenosine was found to reduce HIV-1 replication nearly three decades ago [73]. Recently, compounds that are highly specific for the activation or inhibition of components of the m^6^A methyltransferase complex have been developed [74,75]. Compounds that activate the METTL3/14/WTAP complex increase HIV-1 production in latently infected cells subject to reactivation [39]. Naturally, the use of compounds that enhance or inhibit m^6^A modification in a clinical setting would raise concerns about pleiotropic effects on the m^6^A modification of cellular RNA. Nevertheless, METTL3 inhibitors have shown promise in the treatment of cancers in animal models and are currently being tested in human clinical trials [75,76,77]. 

It will first be important to determine whether and how m^6^A regulates HIV-1 replication in infected individuals. A study of gene expression in peripheral blood mononuclear cells (PBMCs) isolated from HIV-1-infected individuals showed a significant correlation between the size of the latent HIV-1 reservoir and the expression of METTL3 [78]. Conversely, there was an inverse correlation between the size of the latent reservoir and the expression of ALKBH5. These data suggest that higher levels of HIV-1 replication may correlate with higher levels of m^6^A in RNA from PBMCs of HIV-1-infected patients. However, m^6^A levels were not measured in this study. An elevation of cellular m^6^A levels in HIV-1-infected individuals would be consistent with the results of *in vitro* and *ex vivo* infections. Future studies using blood or tissue CD4+ T cells collected from HIV-1 patients with viremia and on suppressive antiretroviral therapy will be useful in determining how m^6^A levels are affected by HIV-1 infection *in vivo*.

## 11. Future Perspectives

Several predicted m^6^A sites in the HIV-1 RNA are in regions overlapping with structural features that are important for virus replication or near functional domains such as SA or NF-kB binding sites [25,26,31,58]. Mutation of these residues can therefore lead to perturbation of RNA secondary or tertiary structures, although structure is often not assessed in these studies. An alternative approach to investigating the role of m^6^A during viral replication is by manipulating the levels of expression of m^6^A writers or erasers in cells. However, this strategy is not sequence-specific and can also lead to indirect effects that are not straightforward to interpret. This can be partially addressed when investigating the role of m^6^A during the pre-integration stages of infection by using viruses produced in cells overexpressing erasers to specifically reduce m^6^A on viral RNA only without perturbing the modification of host cell RNA [48,79]. However, this approach also does not allow for the specificity required for investigating individual m^6^A functions. Nevertheless, in the absence of robust techniques allowing for better specificity, many groups have investigated how components of the m^6^A methyltransferase complex and m^6^A demethylases influence HIV-1 replication.

With rapid advancements in the technology for studying the role of epitranscriptomic modifications, mechanistic studies can be enhanced. Many m^6^A sequencing approaches have been developed that now allow for the identification of m^6^A at a single-nucleotide resolution, which is a vast improvement upon original meRIP-seq and even PA-m^6^A-seq approaches [80,81,82,83]. In addition, CRISPR/Cas9 technology has been harnessed for the sequence-specific targeting of adenosines of interest for the addition or removal of methyl groups [20,21,22,23,24,84]. Whether these techniques are robust enough to be of use for the delineation of m^6^A-dependent phenotypes in the context of viral infections remains to be confirmed. Regardless, these more powerful tools can be harnessed to explore the molecular mechanisms by which m^6^A modification of HIV-1 RNA regulates various phases of the viral life cycle. These mechanistic studies will benefit our understanding of HIV-1 replication and viral persistence.

## Figures and Tables

**Figure 2 viruses-16-00127-f002:**
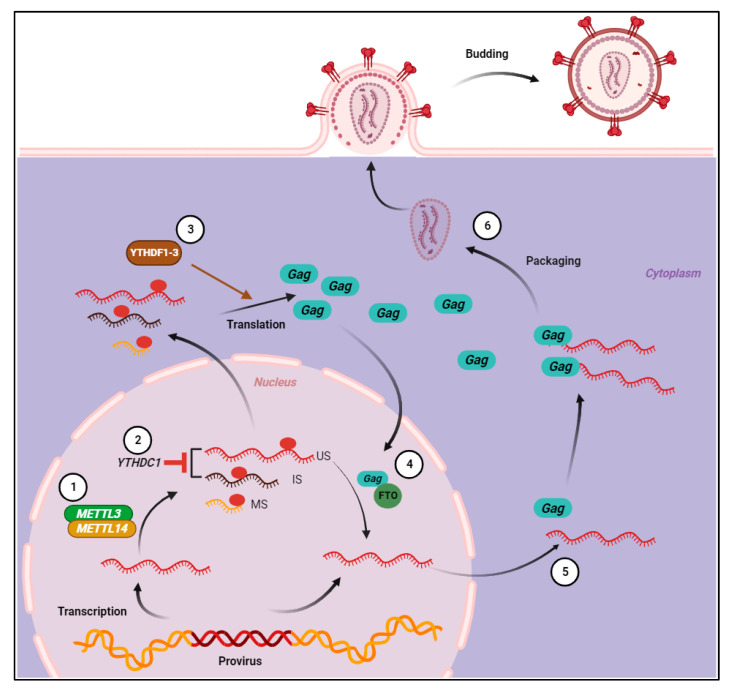
Post-integration regulation of HIV-1 RNA by m^6^A regulators. (1) Viral RNA is methylated by the methyltransferase complex in the nucleus (for simplicity, only one m^6^A is shown on viral transcripts). (2) YTHDC1 regulates the splicing of full-length and incompletely spliced HIV-1 RNA but does not affect multiply spliced transcripts. (3) YTHDF1, 2, and 3 proteins positively regulate HIV-1 RNA abundance and viral protein synthesis. (4) Gag is imported into the nucleus and interacts with FTO to demethylate full-length HIV-1 RNA. (5 and 6) Gag preferentially interacts with full-length HIV-1 RNA and is packaged into progeny virions.

**Table 1 viruses-16-00127-t001:** Methods for mapping m^6^A modification sites of HIV-1 RNA.

HIV-1 Strain	Cell Type	Sequencing Method	References
LAI	MT4	meRIP-seq ^3^	[25]
NL4-3 ΔEnv VSV g ^1^	CEM	PA-m^6^A-seq	[26]
NL4-3	Jurkat Primary CD4+ T cells	meRIP-seq	[27]
NL4-3	CEM	PA-m^6^A-seq ^4^	[28]
NL4-3 GFP ΔEnv VSV g ^2^	SupT1	meRIP-seq	[29]
NL4-3 ΔEnv VSV g	HEK293T	meRIP-seq	[30]
NL4-3	HEK293T	Direct RNA sequencing	[31]

^1^ Single-cycle VSV g pseudotyped HIV-1_NL4-3_; ^2^ Single-cycle VSV g pseudotyped HIV-1_NL4-3_ with a GFP reporter; ^3^ Methylated RNA immunoprecipitation sequencing; ^4^ Photo-crosslinking-assisted m^6^A sequencing.

## Data Availability

Not applicable.

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
