# Peer review of "Functional Impacts of Epitranscriptomic m6A Modification on HIV-1 Infection"

_viruses, 2024, doi:10.3390/v16010127_

Round 1
Reviewer 1 Report
Comments and Suggestions for Authors
Authors performed a very good analysis of the literature that nicely summarizes our current understanding of the role of the RNA modification m6A during HIV-1 replication. I have no comments and suggest the acceptance of the manuscript for publication.
Congratulations to the authors!
Author Response
We appreciate this reviewer's positive and encouraging review. There are no specific comments to be addressed.
Reviewer 2 Report
Comments and Suggestions for Authors
This is a very well written review. I especially appreciate the discussion of caveats, that it is important to consider differences between studies like HIV strain, time course, cell type, etc. My minor comments are below:
1. Considering many of the studies discussed are in T cells, and there is not an incredibly large focus on innate immunity, perhaps reconsider the manuscript title or include more content/create a larger focus on innate immunity.
2. Mention R5/X4 tropism earlier in the manuscript where there is the discussion of HIV strains (top of page 4). Saying specifically which strains are R5 or X4 better drive home the point the authors are making there.
3. A more thorough discussion of myeloid vs lymphoid HIV infection would be helpful to better appreciate why it is so important to understand similarities/differences in m6a regulation in these cell types (as they play different roles in the course of infection).
4. Is anything known about m6a regulation and viral reservoirs that could be discussed?
Author Response
See attached responses to reviewer 2.
